# Exploring the impact of drug decriminalization and legalization policies on mental health outcomes: A scoping review

**Mana Mohebbian**[1,2]*, **Sara Najafi**[3], **You Na Choi**[1,2], **Christian Schütz**[2,4], **Rosemin Kassam**[1], **Arminee Kazanjian**[1], **Joseph Puyat**[1,2]

**1** School of Population and Public Health, University of British Columbia, Vancouver, British Colombia, Canada, **2** Center for Advancing Health Outcomes, Providence Health Care, Vancouver, British Colombia, Canada, **3** Nova Scotia Health Authority, Halifax, Nova Scotia, Canada, **4** Department of Psychiatry, University of British Columbia, Vancouver, British Colombia, Canada

* mana1994@student.ubc.ca

## Abstract

As countries increasingly adopt more liberal drug policies, concerns have emerged about their broader health and social impacts. A complex bidirectional relationship exists between problematic drug use and mental health conditions. This is particularly evident in the co-occurrence of mental health disorders with substance use disorders (SUDs). However, the broader mental health effects of drug policy remain underexplored. This review aims to map existing research on non-SUD mental health outcomes associated with drug decriminalization and legalization policies. We conducted a scoping review following JBI guidelines and the PRISMA-ScR checklist. Studies published between January 2001 and December 2024 were included if they examined non-SUD mental health outcomes related to drug policy changes, with a focus on decriminalization, legalization, or commercialization. We searched Medline, EMBASE, CINAHL, PsycInfo, and Web of Science, and manually screened relevant policy reports. Only English-language studies were included. Data extraction and analysis were conducted using Covidence, with a descriptive summary of study characteristics and findings. A total of 55 studies met inclusion criteria, comprising 16 review papers and 39 original research articles (37 quantitative and 2 qualitative). Most evidence came from the United States (n = 29) and Canada (n = 18). No studies examined the mental health impacts of non-cannabis drug policies or decriminalization frameworks. The most frequently assessed outcomes were psychosis, suicide, and depression. This review maps the current evidence base and identifies major gaps, especially concerning decriminalization and policies targeting substances other than cannabis. The heterogeneity in study designs and policy contexts highlights the need for multi-faceted, context-sensitive research to inform future policy.

**Data availability statement:** All relevant data are included and presented within the manuscript.

**Funding:** The authors received no specific funding for this work.

**Competing interests:** The authors have read the journal's policy and have the following competing interests: Dr. Christian Schütz holds a partner academic appointment with the University of British Columbia (UBC), supported by funding from the Provincial Health Services Authority (PHSA). He also works as a psychiatrist and clinician scientist at PHSA and serves on the advisory board of Clearmind, a startup focused on medication development. There are no patents, products in development, or marketed products associated with this research to declare. This does not alter our adherence to PLOS ONE policies on sharing data and materials.

## Introduction

In recent years, global drug policy has undergone a significant shift, moving away from punitive, prohibitionist approaches toward more reform-oriented models. These "liberal drug policies"—commonly defined as regulatory frameworks that include decriminalization, legalization, and commercialization—aim to reduce criminal penalties, regulate markets, and prioritize public health. To date, over 30 countries have implemented some form of decriminalization to lessen the legal consequences associated with personal drug use [1]. In 2018, the United Nations guidance has emphasized the importance of aligning drug policy with public health, development, and human rights objectives, noting that countries should "place people, health and human rights at the center" of drug policy responses [2].

Despite this momentum, concerns persist about potential increases in substance use and associated harms, particularly among vulnerable populations such as youth [3]. While punitive policies have largely been shown to be ineffective in reducing drug-related harm [4], the broader public health effects of liberalization remain contested.

Drug policies, including decriminalization and legalization, offer different methods of moving away from traditional prohibitionist frameworks. Decriminalization generally involves reducing or eliminating penalties for the possession of small amounts of controlled substances, focusing on personal use while keeping production and sale illegal [5]. On the other hand, legalization creates a regulated market for certain drugs, allowing legal production, sale, and use.

Opponents argue that by increasing the availability of these substances, legalization may lead to higher consumption and, consequently, a rise in negative health outcomes. Advocates, however, primarily emphasize the public health benefits of legalization and decriminalization, arguing that these policies can save lives by preventing overdose deaths, and through supervised consumption and regulated supply, expanding harm-reduction and treatment services, and avoiding incarceration-related trauma by shifting from criminalization to care [6]. They also note that legalization enhances control over drug quality, disrupts illegal markets, and generates tax revenue that can be reinvested into public health initiatives [7,8]. Mainstream narratives, often shaped by non-expert voices such as politicians, media figures, and ideologically driven commentators, frequently reflect polarized views. The absence of conclusive evidence, combined with limited public awareness, has further fueled misperceptions about the policy impacts [9,10].

Drug-related outcomes—whether beneficial or harmful—are highly dependent on contextual factors such as health system capacity, enforcement practices, public attitudes, and the pre-existing status of drug use, treatment infrastructure, and criminal justice involvement within each country or jurisdiction. For example, Portugal's decriminalization efforts have reduced the burden on criminal justice systems and improved public health outcomes by increasing access to harm reduction services, although concerns about potential increases in drug use persist [5]. Similarly, the Netherlands' depenalization of cannabis through "coffee shops" has lowered criminal

justice costs but has also been associated with higher cannabis use, driven more by commercialization than decriminalization itself [11]. In Uruguay, the state-controlled cannabis market aimed to regulate the illegal drug trade, though its impact on overall use is still debated [12]. In Canada, cannabis legalization has reduced cannabis-related arrests and provided social justice benefits but has also led to increased emergency department visits and hospitalizations, particularly among youth [13,14].

In contrast, countries such as France and Sweden, which maintain prohibitionist drug policies, have seen persistent or increasing rates of drug use and related harms, suggesting that restrictive laws alone may not effectively curb substance use [15,16]. Japan, on the other hand, also enforces a highly punitive drug policy, but reports low levels of drug use [17]. These contrasts suggests that the effectiveness of drug policy cannot be evaluated in isolation from the broader social, health, and legal context in which it is implemented.

Beyond legal and economic effects, liberal drug policies may have profound implications for public mental health. Substance use and mental health share a complex, bidirectional relationship: each can precipitate or aggravate the other [18]. Importantly, poor mental health, trauma, and adverse childhood experiences (ACEs) are recognized as significant upstream drivers of substance use initiation and escalation [19]. Individuals exposed to ACEs, untreated psychological conditions, or chronic stress are at substantially higher risk of developing substance use disorders (SUDs) and related harms [20]. At the same time, problematic drug use can contribute to a range of psychiatric symptoms, including depression, anxiety, and psychosis [18]. Even occasional or nondependent consumption may provoke mental health conditions [21], yet these co-occurring complications are particularly prevalent among individuals with SUDs [22].

The nature and severity of these mental health effects depend on the type of substance, the pattern of use, and individual vulnerability. For example, early and heavy cannabis use in adolescence can disrupt neural maturation and heighten the risk of psychosis, depression, and suicidal ideation [23]. Problematic use of opioids, whether prescribed or illicit, are linked to depression, anxiety, and suicidality, especially during withdrawal phases [24,25]. Stimulants such as cocaine and amphetamines may trigger acute paranoia and psychotic episodes, with chronic use often resulting in persistent cognitive deficits [26,27]. Harmful use of legally available substances like alcohol can also elevate the risks of depression and suicide [28]. These overlapping conditions not only complicate diagnosis and treatment but also point to shared biological and psychosocial vulnerabilities.

Despite strong associations, it remains uncertain whether escalating substance use worldwide is contributing to mental health conditions in the context of substance use or whether worsening psychological well-being is driving more individuals to self-medicate with substances. In this context, evaluating how drug policy, be it legalization, decriminalization, or harm-reduction strategies, modulates these intertwined outcomes is an essential focus for future research.

While the issue is pressing, studies exploring the mental health impact of changing drug policies remain sparse. A recent scoping review by Fortier et al. investigated the mental-health impacts of cannabis legalization [29], identifying twenty-eight original studies and reports from January 1, 2012, to April 30, 2023. This paper has outlined the limited research available and highlighted the largely inconclusive findings, however, its focus was specifically on cannabis policies.

The present review expands the scope to include a broader range of drug policies, such as the decriminalization, legalization or commercialization of psychoactive substances other than cannabis, alcohol, tobacco, or prescribed medications, and extends the analysis period back to January 2001. We also incorporate review papers alongside original studies to map the entire body of existing literature. This approach offers a more comprehensive, up-to-date overview of current knowledge. By addressing a wider spectrum of policies and substances, this review aims to bridge existing gaps, establish a stronger foundation for future research, and better inform evidence-based policy decisions in this evolving and critical field.

## Methods

We conducted a scoping review to provide an overview of the global literature on the mental health outcomes of drug decriminalization and legalization policies. We adhered to the Joanna Briggs Institute (JBI) guidelines for scoping reviews (Chapter 10) [30] and used the Preferred Reporting Items for Systematic Reviews and Meta-Analyses for Scoping Reviews (PRISMA-ScR) checklist to guide our reporting [31]. A review protocol was developed that outlined our search strategy, eligibility criteria, and the key study features to be extracted.

### Search strategy

We executed the initial search on November 30th, 2023 and then ran a second search on December 21st, 2024 to update our results and capture the most up-to-date evidence. We searched through MEDLINE (Ovid), Embase (Ovid), CINAHL, PsycInfo (EBSCO), Science Citation Index and Social Sciences Citation Index (Web of Science). Government and national institution websites such as EMCDDA and UNODC (United Nations Office on Drugs and Crime) were explored manually for working papers and reports. A search strategy was developed and tested with the help of a professional librarian. We used search terms that include any relevant clusters of terms related to recreational or illicit drug categories, such as "cannabis", "opioid", "stimulant" or "psychedelic", in combination with keywords pertaining to policy regulations, including "decriminalization", "legalization", "liberalization" or "commercialization". We also included MeSH terms for some specific concurrent disorders in the context of substance use. A copy of the full draft of the Ovid EMBASE search strategy can be found in the Supporting information (S1 Appendix).

### Exclusion and inclusion criteria

**Publication type.** Eligible studies included 1) original research employing qualitative, quantitative, or mixed-methods approaches, 2) review papers such as systematic reviews, meta-analyses, narrative reviews, and scoping reviews, and 3) policy reports and working or white papers that had not undergone formal peer review were also considered. However, we did not include commentaries, letters to editors, books, dissertations, theses, protocols, formative research, and conference papers. Studies published before January 1st, 2001 and those not written in English were also excluded.

**Population.** The review included studies involving adults aged 18 years and older, as drug use or possession typically remains illegal for youth and adolescents, even under liberal policy frameworks. Studies focusing on adult sub-populations, such as people who use drugs, individuals with mental health conditions, and those with chronic conditions, were considered in this review.

**Intervention.** Our initial aim was to assess the effects of liberal policies targeting non-cannabis psychoactive substances (e.g., opioids, stimulants) at national or jurisdictional level. Here, the "policy framework" encompasses three main regulatory approaches:

- Decriminalization: removal of criminal penalties for possession or personal use, typically replacing them with administrative fines or diversion to treatment [32].

- Legalization: establishment of a lawful pathway for production, distribution, and consumption under government oversight [33].

- Commercialization: introduction of a regulated market in which private entities can produce, market, and sell substances, often alongside taxation and advertising controls [34].

Because our targeted search for studies on substances other than cannabis yielded no eligible papers, we broadened our scope to include evaluations of cannabis policies. Consequently, any study examining the impact of decriminalization, legalization, commercialization, or similar regulatory changes was considered eligible, regardless of substance class.

**Outcome.** The primary outcome of interest was non-SUD mental health conditions such as depression or anxiety disorders among populations with exposure to psychoactive substances, regardless of whether the individual meets criteria for a substance use disorder. These conditions may be pre-existing, co-occurring, precipitated, or influenced by substance use. This includes both acute mental health conditions triggered by substance use (e.g., psychotic episodes during intoxication) and longer-lasting or recurrent disorders that may persist beyond the immediate effects of the substance [35,36]. We included studies reporting population-level measures derived from national statistics, administrative health records, or large cohort studies. Both large, population-based datasets and smaller clinical or community samples were accepted, provided they met our diagnostic or measurement criteria. Studies were included if they employed clinical diagnoses (based on DSM or ICD criteria), validated diagnostic interviews, prescription-fill records for major psychiatric medications, or standardized self-report scales to ascertain mental health outcomes. We excluded any studies that:

- Reported solely on patterns of drug consumption (e.g., frequency, quantity) or on problematic use indicators (e.g., dependence, abuse) or all-cause drug-related emergency department visits (e.g., overdose rates, drug-related deaths, intoxication, or withdrawal) without measuring psychiatric symptoms;

- Focused exclusively on substance use disorders (SUDs). While SUDs are recognized substance-related mental health conditions, they were excluded to allow a focused examination of non-SUD mental health outcomes in the context of substance use, such as depression, anxiety, or psychosis;

- Investigated physical health outcomes, civic and social impacts, motor vehicle accidents, drug-related arrests, law enforcement activity, or any aspect of drug trafficking, manufacturing, or cultivation;

- Addressed mental health conditions associated exclusively with prescribed medications, alcohol, tobacco, or caffeine. Prescribed cannabis and psychedelics were treated as distinct exposures, as their medical legalization falls within the scope of this review. Here "prescribed medications" refers specifically to psychoactive substances commonly used without policy-specific regulation (e.g., benzodiazepines).

### Selection process and data charting

The search results were imported into the Covidence platform for screening, full-text review, and data extraction [37]. Titles and abstracts were screened independently by two reviewers, with conflicts resolved by a third reviewer. All potentially eligible studies underwent a full-text review by two reviewers, and any disagreements during this stage were resolved through discussion.

Data extraction and charting focused on key study features and findings, including: (1) study design, (2) country and region of the study, (3) data sources, (4) type of drug-related policy (e.g., decriminalization, legalization, commercialization), (5) relevant outcomes, and (6) reported findings. A copy of the template data charting tool is available in the Supporting Information (S2 Appendix).

A descriptive summary was synthesized to provide an overview of the characteristics, distribution, and findings of the studies. To avoid overstating results, we charted the characteristics and findings of original studies—including quasi-experimental, cross-sectional and cohort studies—separately from review articles.

## Results

### Review process and PRISMA

In total, 5,607 studies were identified through both searches (4,805 in November 2023 and 802 in December 2024). The breakdown of sources is as follows: Medline (n = 1,191), Embase (n = 2,956), PsycINFO (n = 576), CINAHL (n = 273), Web of Science (n = 605), and manual sources (n = 6). After removing duplicates, 3,373 studies remained for title and abstract

screening. Of these, 3,215 ineligible studies were excluded, leaving 158 full texts for review. At the full-text review stage, 103 studies were further excluded. This resulted in a total of 55 studies eligible for data extraction and charting. The most frequent reason for exclusion at this stage was the lack of reported mental health measures as an outcome. Fig 1 presents the PRISMA flow diagram outlining the multiple steps of this scoping review. The PRISMA-ScR checklist is provided in the Supporting Information (S3 Appendix).

## Descriptive summary

Of the 55 included studies, 16 were review papers and 39 were original research papers. No policy reports or working papers were identified that addressed mental health outcomes. The characteristics of the review and original research papers are described in greater detail in the following sections. Nearly all studies originated from North America, with 29 conducted in the United States and 18 in Canada. One study was conducted in the United Kingdom, and seven review papers took a global perspective, although their investigators were predominantly based in North America. Fig 2 illustrates the geographical distribution of the assessed studies.

We could not locate any studies examining the impact of policies decriminalizing or legalizing 'non-cannabis psychoactive drugs', including substances other than cannabis, alcohol and tobacco, on mental health outcomes (n = 0). Consequently, all included studies were focused on cannabis policies (n = 55). In terms of mental health outcomes, self-harm (including suicidal ideation, suicide deaths, and intentional self-harm) was the most frequently assessed outcome (n = 22), followed by psychosis (n = 21), depression (n = 11), anxiety (n = 9), and psychiatric ED visit/hospitalization (n = 11), as shown in Fig 3. Tables 1 and 2 present the characteristics of the original studies and review papers and their findings.

## Time trends and policy frameworks

Our search covered a 24-year period from January 2001 through the end of 2024. However, as shown in Fig 4, all included studies were published within the past 10 years, with none identified prior to 2014. This highlights a notable gap in the literature, especially considering that cannabis decriminalization in the United States began as early as the 1970s, followed by the legalization of medical cannabis in California in 1996 [92] and the introduction of medical cannabis legislation in Canada in 2001 [93].

Among the 55 reviewed papers, 29 evaluated the effects of recreational legalization, with or without including the impact of opening commercial dispensaries; 7 focused only on medical legalization; and 18 analyzed the impacts of both medical and recreational laws. Thirteen studies considered commercialization and opening of the cannabis dispensaries in their analyses. One review paper only examined the literature on commercialization and access to cannabis retailers [79]. Notably, none of the original studies or review papers directly examined decriminalization as a distinct drug policy framework.

## Summary of original studies

A majority of original studies took place in the United States (n = 24) [38–40,42,44,47–51,53–55,59–61,63,64,68–71,73,76] and fourteen studies were conducted in Canada [41,45,46,52,56–58,62,65–67,72,74,75]. We identified one study from the United Kingdom, where a liberal drug policy is not broadly implemented and only allows restricted cannabis prescriptions for medical purposes [43]. For the US studies, analyzed data was obtained from the following states: Colorado (n = 6) [47,48,54,60,71,73], California (n = 4) [40,42,51,60], Washington (n = 3) [47,48,60], Alaska (n = 2) [47,60], Oregon (n = 2) [47,60], Massachusetts (n = 1) [60] and Nevada (n = 1) [60]. The remaining studies in the US incorporated data from nearly all states. In Canada, Ontario was the province predominantly studied (n = 10) [41,45,46,56,57,62,65–67,74], followed by Alberta (n = 4) [45,46,66,75], Quebec (n = 3) [58,66,72], and British Columbia (n = 2) [52,66]. Most studies employed a quasi-experimental(n = 14) [38,39,42,45–48,53,57,60,63,70,75,76] or a cohort design (n = 12) [40,41,49–52,58,61,65,71–73]. Nine

Scoping review - Exploring the impact of drug decriminalization and legalization policies on mental health outcomes

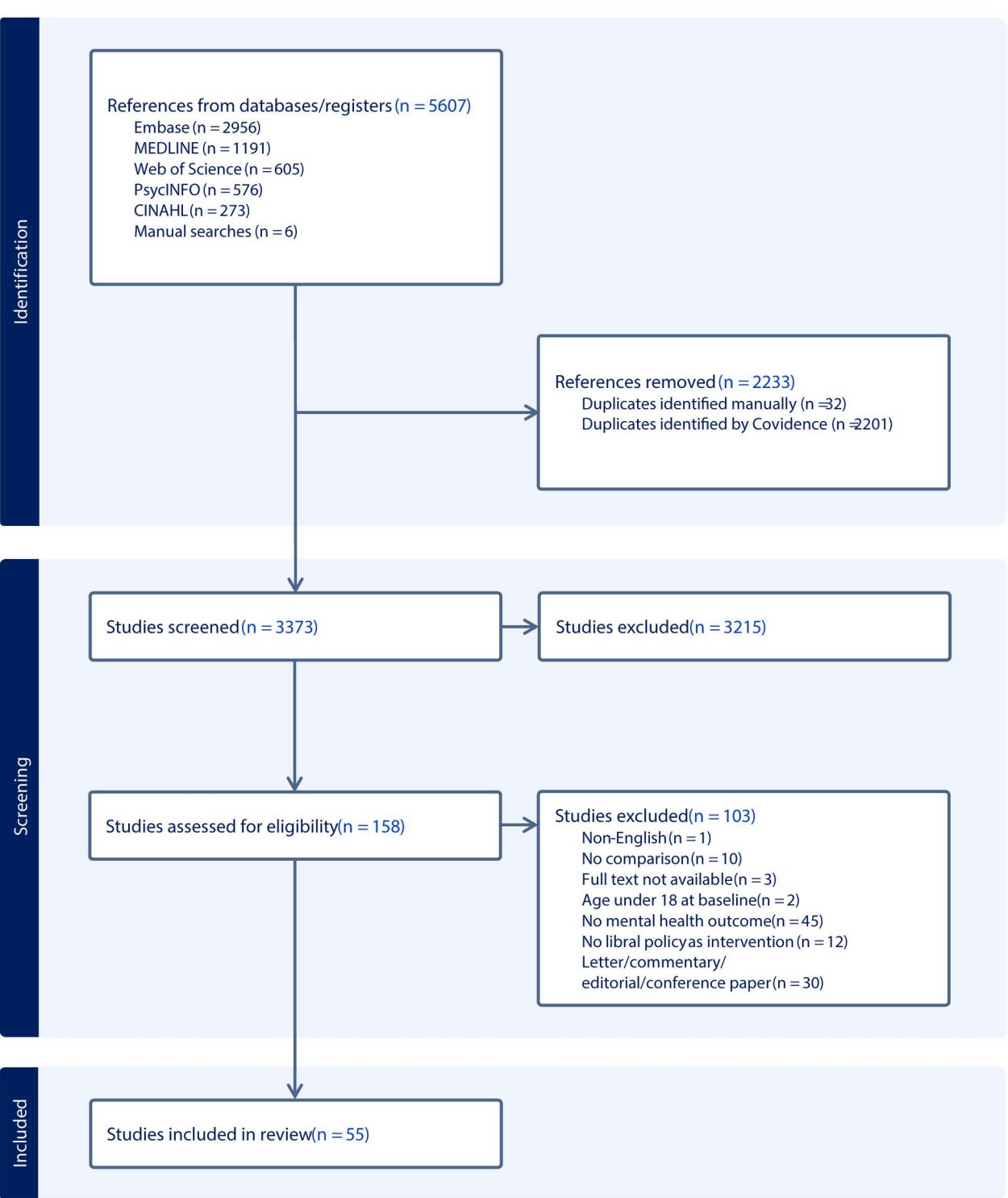

**Fig 1. Preferred Reporting Items for Systematic Reviews and Meta-analyses extension for scoping review (PRISMA-ScR) flow diagram.**

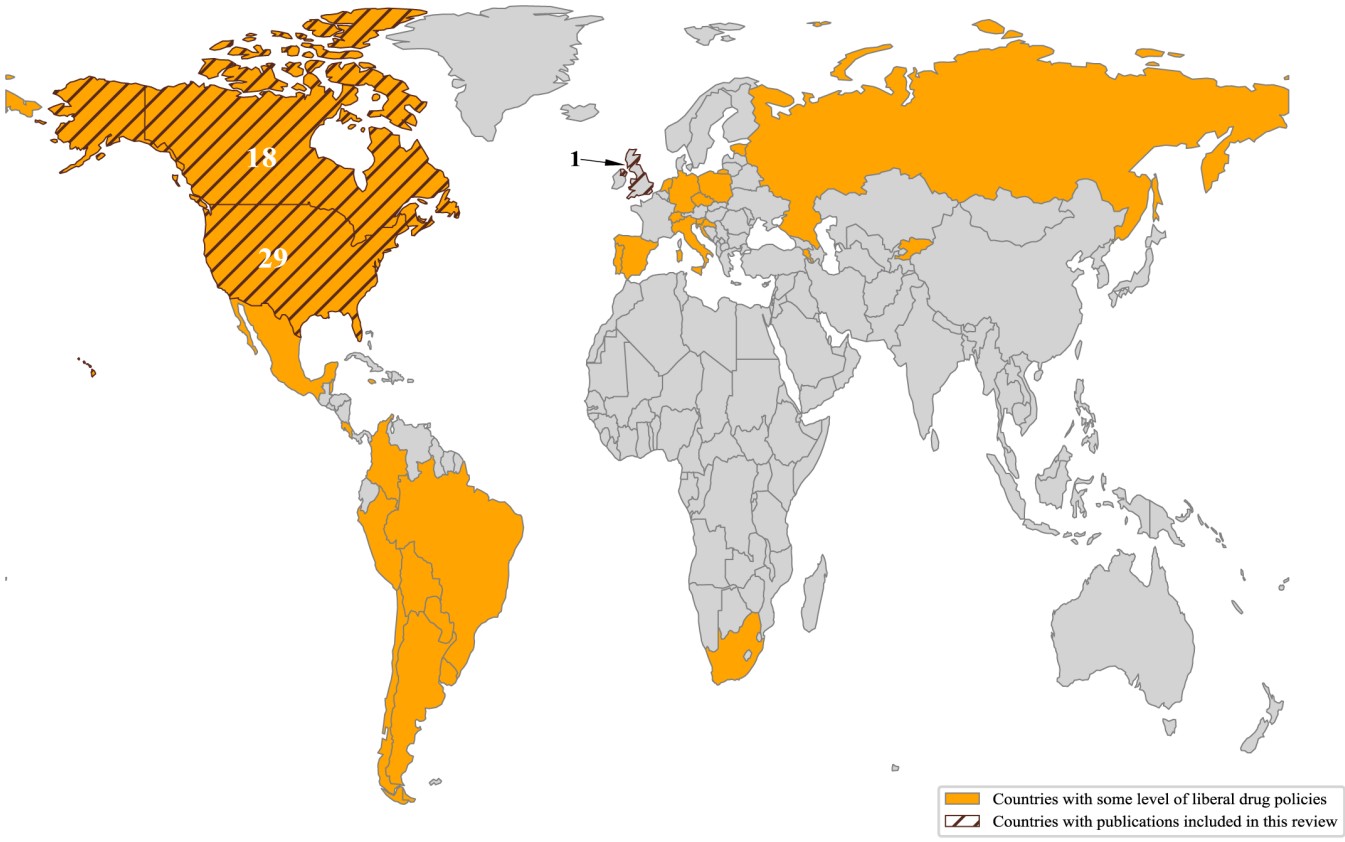

**Fig 2. World map demonstrating the countries that have implemented some level of liberal drug policies versus countries that had publications included in this scoping review.** Data on countries' policies were collected from Eastwood et al. [1]. The figure is generated using Python (Base layer source: Natural Earth, 1:110m Cultural Vectors (Admin 0 – Countries), available at https://www.naturalearthdata.com/downloads/110m-cultural-vectors/).

were cross-sectional studies [44,54–56,59,62,64,66,67], including those with repeated cross-sectional design. One study used an ecological cohort approach [69], and another referred to their methodology as an event study [68]. There were two qualitative studies in which data were collected through semi-structured interviews: one with healthcare providers in a psychiatric hospital setting [74], and the other with individuals who had a prescription for medical cannabis and their carers [43].

Most papers utilized data from national or regional administrative databases, such as vital statistics, emergency department presentations, hospital admission/discharge records, or insurance claims. A few studies relied on previously collected surveys or existing longitudinal cohort data [40,49,63,70,76]. The majority of studies (n = 24) compared pre- and post-legalization trends, primarily using quasi-experimental (e.g., interrupted time series) and longitudinal designs. Fifteen studies compared states or regions that adopted the policy with those that did not, either alongside a pre-post approach or as a standalone comparison. Two studies constructed synthetic control groups to represent counterfactual scenarios in which the policy had never been implemented [42,48]. Additionally, two studies used conditions or emergency department visits unrelated to cannabis as comparison outcomes [45,54].

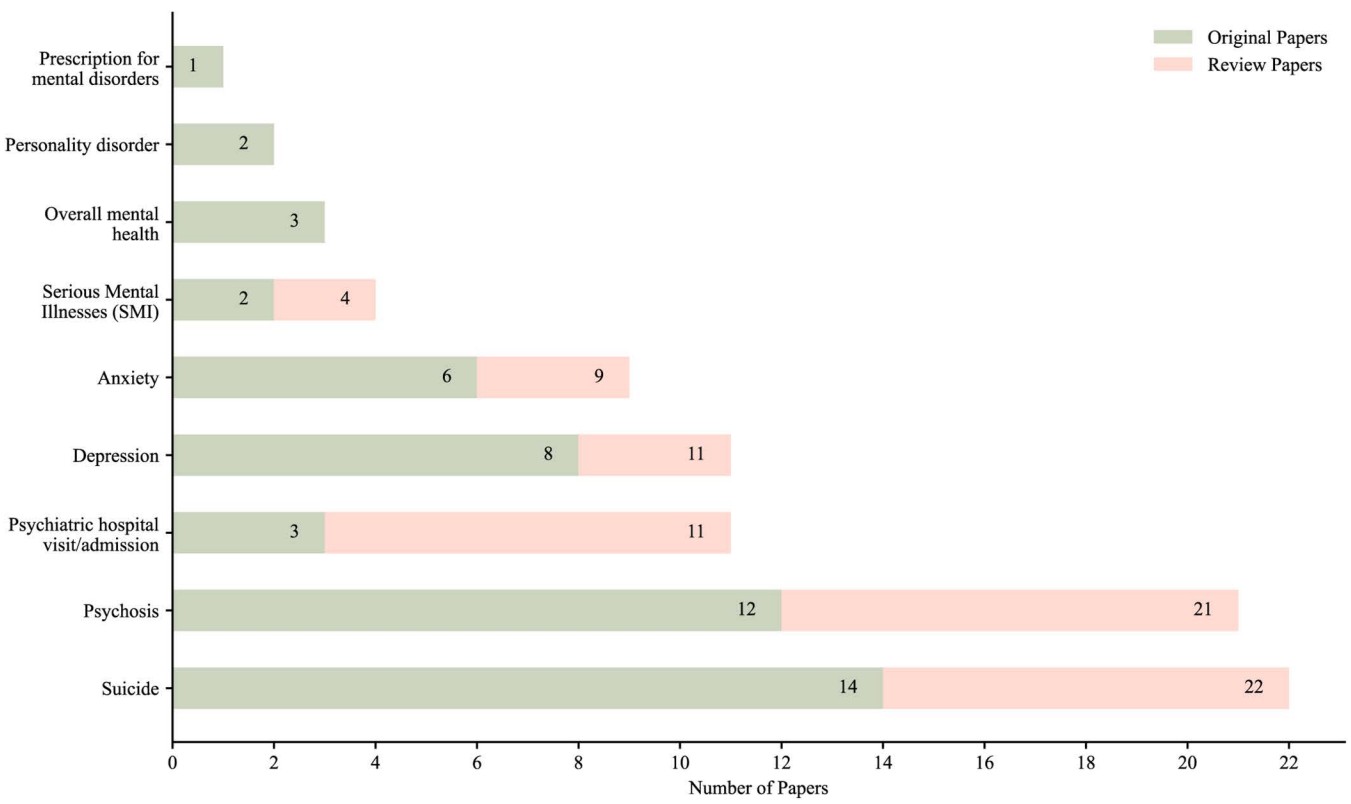

**Fig 3. Number of original and review papers in terms of non-SUD mental health outcomes.**

Suicide and self-harm (including suicide deaths [38,42,47,48,53,55,59,60,70,71], intentional self-harm [46,48,61,67], and suicidal ideation [69]), along with psychosis [41,45,50,57,58,64–66,72,73,75,76] and depression [40,41,51,52,56,63,69,75] were the most frequently evaluated non-SUD mental health outcomes. These were followed by anxiety [40,41,52,56,62,75], overall mental health [39,40,43,70], non-specified psychiatric admissions/visits [54,57,68], serious mental illness (SMI) [49,69], personality disorder [73,75] and somatization [40], as shown in Fig 3. One study analyzed prescription fill rates for psychiatric medications as an indicator of trends in psychiatric symptomatology [44].

It is worthwhile noting that the measures used to assess outcomes varied across studies. Some used more reliable diagnostic methods, such as ICD codes and DSM-based interviews, while others relied on self-report instruments [39,40,52]. One study used prescription fill data as a proxy indicator, which may be influenced by factors unrelated to policy changes and should therefore be interpreted with caution when evaluating mental health impacts.

## Summary of review papers

Table 2 presents a summary of the included review papers. Sixteen review papers were included, comprising nine narrative reviews [78,80,81,84–87,89,91], six systematic reviews [77,79,82,83,88,90] and one scoping review [29]. Half of the reviews looked over papers published globally [29,77,79,81,82,88,90], one third specifically focused on the United States [80,84,85,87,91], and the remaining reviewed the policy impact in Canada [78,83,86,89]. Notably, 20 of the original studies included in our scoping review were also examined in one or more of the included review papers. This overlap reflects the limited pool of available evidence in this area and has been specified in Table 2 to acknowledge repetition and ensure transparency in the interpretation of findings.

| Lead author | Year of publication | Country | Province(s)/state(s) | Policy framework | Study design | Comparison/control group | Sample size | Data source | Outcome | Findings | Overall impact[a] |
|---|---|---|---|---|---|---|---|---|---|---|---|
| Anderson [38] | 2014 | United States | Nationwide | ML | Quasi-experimental study | States which did not adopt drug policy | 918 state-year observations | Mortality Detail Files (produced by the National Vital Statistics System) | Suicide rates | A significant reduction in the rate of suicide aiming 20–39 yrs-old men following medical cannabis legalization. | Positive |
| Andreyeva [39] | 2019 | United States | Nationwide | ML | Quasi-experimental study | States which did not adopt drug policy | Adopted MML [n=2,312,647]; Did not adopt MML [n=2,683,474] | Behavioral Risk Factor Surveillance System (BRFSS) | Days with mental health problems | Individuals reported fewer days with mental health issues in states with medical legalization, while dispensaries alone showed no effect. The combined impact of legalization and dispensaries offered weak evidence of mental health improvement. | Positive |
| Ataiants [40] | 2024 | United States | California | ML RL | Cohort study | Pre-RCL | Baseline [n= 366]; Wave 6 [n= 251] | Cannabis, Health, and Young Adults (CHAYA) project data | Overall psychological distress Anxiety Depression Somatization | No significant differences in overall psychological distress, anxiety, or depression, throughout the study period, which includes the transition from medical-only to adult-use legal cannabis. | No impact[b] |
| Baraniecki [41] | 2021 | Canada | Ontario | RL | Cohort study (Retrospective) | None | 358 | Discharge Abstract Database Epic medical record system at St. Joseph's Healthcare Hamilton | ED psychiatric visits with the following symptoms: bizarre behaviour depression hallucinations/delusions anxiety/crisis | No evidence of association between legalization and rate of ED visits with cannabis-related psychiatric symptoms. | No impact |

*(Continued)*

Table 1. (Continued)

| Lead author | Year of publication | Country | Province(s)/ state(s) | Policy framework | Study design | Comparison/ control group | Sample size | Data source | Outcome | Findings | Overall impact[a] |
|---|---|---|---|---|---|---|---|---|---|---|---|
| Bartos [42] | 2020 | United States | California | ML | Quasi-experimental study | A synthetic control representing the scenario in which the policy had not been enacted | N/A | National Center for Health Statistics | Total Suicide rate Gun suicide rate Non-gun suicide rate | Rates of both total suicide and suicide involving a gun, decreased significantly after medical legalization. The impact on non-gun suicides was non-significant. | Positive |
| Beckett-Wilson [43] | 2023 | United Kingdom | Not specified | ML | Qualitative study | None | 24 | Individual interviews | Mental health issues | Patients with mental illnesses (including PTSD, depression and anxiety) experienced improvements in their mental health after legalization of medical cannabis. | Positive |
| Bradford [44] | 2024 | United States | Nation-wide | ML RL COM | Cross sectional study | Pre-RCL Pre-MCL | 10,013,948 | Optum's deindentified Clinformatics Data Mart Database | Mental health disorder prescriptions fill for medication classes: (1) benzodiazepines, (2) antidepressants, (3) antipsychotics, (4), barbiturates, and (5) sleep medications | Both ML and RL were associated with a reduction in benzodiazepine prescription fills, while an increase in the dispensing of antipsychotics and antidepressants was observed. | Mixed |
| Callaghan [45] | 2022 | Canada | Ontario Alberta | RL | Quasi-experimental study | Pre-RCL Amphetamine- and alcohol-psychosis | Cannabis-induced psychosis [n=5,832]; Schizophrenia and related conditions [n=211,661]; Amphetamine-related psychosis [n=10,829] | National Ambulatory Care Reporting System (NACRS) | Cannabis-induced psychosis Schizophrenia and related disorders Amphetamine-induced psychosis | No significant evidence of change in the rate of ED visits for cannabis-induced psychosis or schizophrenia 14 months after recreational legalization in Canada. | No impact |

*(Continued)*

| Lead author | Year of publication | Country | Province(s)/ state(s) | Policy framework | Study design | Comparison/ control group | Sample size | Data source | Outcome | Findings | Overall impact[a] |
|---|---|---|---|---|---|---|---|---|---|---|---|
| Cusimano [46] | 2023 | Canada | Alberta Ontario | RL | Quasi-experimental study | Pre-RCL | N/A | National Ambulatory Care Reporting System (NACRS) Discharge Abstract Database (DAD) | ED visits for intentional self-harm Purposely self-inflicted poisoning or injury, including attempted suicide Hospitalizations for intentional self-harm | No significant evidence of change in the rate of ED visits for intentional self-harm (including attempted suicide) after recreational legalization. | No impact |
| Dills [47] | 2016 | United States | Colorado Washington Oregon Alaska | RL | Quasi-experimental study | Pre-RCL | N/A | Centers for Disease Control and Prevention, CDC Wonder Portal | Annual suicide rate | No significant evidence of change in suicide rates following recreational legalization. | No impact |
| Doucette [48] | 2021 | United States | Washington Colorado | RL COM | Quasi-experimental study | A synthetic control representing the scenario in which the policy had not been enacted. | N/A | National Center for Health Statistics (NCHS) | Death by suicide intentional self-harm events | A significant increase in death by suicide among 15–24 year-olds in Washington after recreational legalization. No significant change in suicide rates other age groups in Washington or in any age group in Colorado. | Mixed[c] |
| Dutra [49] | 2018 | United States | Nationwide | ML | Cohort study | States which did not adopt the drug policy | 306 state-year observations | National Survey on Drug Use and Health (NSDUH) | Serious mental illness (SMI) | Higher prevalence of SMI in states with liberal medical legalization | Negative |
| Elser [50] | 2023 | United States | Nationwide | ML RL COM | Cohort study | States which did not adopt the drug policy | 63,680,589 | Optum Clinformatics Data Mart Database | Psychosis-related diagnoses Prescribed antipsychotics | No significant difference in rates of psychosis-related diagnoses and prescription of antipsychotics in states with medical or recreational legalization (with or without commercialization) | No impact |

*(Continued)*

Table 1. (Continued)

| Lead author | Year of publication | Country | Province(s)/state(s) | Policy framework | Study design | Comparison/control group | Sample size | Data source | Outcome | Findings | Overall impact[a] |
|---|---|---|---|---|---|---|---|---|---|---|---|
| Gali [51] | 2021 | United States | California | RL | Cohort study | Pre-RCL | 429 | Stanford University's WELL for Life registry | Depression diagnosis | Having another mental health illness diagnosis is correlated with cannabis use after recreational legalization | Negative |
| Geoffrion [52] | 2021 | Canada | British Columbia | RL | Cohort study (Retrospective) | Pre-RCL | 509 (pre-legalization [n = 366]; post-legalization [n = 143]) | Prospective registry at a tertiary care referral clinic for pelvic pain and endometriosis (EPPIC) | Depression Anxiety | Significantly higher anxiety scores in post-legalization cannabis users. Higher but non-significant depression scores in post-legalization cannabis users. | Mixed |
| Grucza [53] | 2015 | United States | Nation-wide | ML | Quasi-experimental study | States which did not adopt drug policy Pre-MCL | 662,993 | Multiple Cause of Death files, collected by the National Center for Health Statistics; National Association for Public Health Statistics and Information Systems (NAPHSIS) | Suicide rates | No significant evidence of association between medical cannabis legalization and suicide death, after adjustment for key covariates. | No impact |
| Hall [54] | 2018 | United States | Colorado | RL COM | Cross sectional study | ED visits not attributable to cannabis | 4,800,644 | Colorado Hospital Association ED (CHAED) discharge data discharges from the University of Colorado Hospital ED (UCHED) | Cannabis-associated diagnostic code and a mental health-primary diagnosis acute psychiatric visit | Significant increase in the prevalence of cannabis-associated ED visits with primary mental health diagnoses surrounding the legalization and commercialization period in Colorado. | Negative |

*(Continued)*

**Table 1.** (Continued)

| Lead author | Year of publication | Country | Province(s)/ state(s) | Policy framework | Study design | Comparison/ control group | Sample size | Data source | Outcome | Findings | Overall impact[a] |
|---|---|---|---|---|---|---|---|---|---|---|---|
| Hammond [55] | 2024 | United States | Nationwide | ML RL | Cross sectional study | States which did not adopt drug policy | 113,512 | National Center for Health Statistics National Vital Statistics System | Deaths by suicide | Female youth residing in ML and RL states faced a 10% and 16% higher risk of suicide, respectively. In contrast, no consistently significant relationships were observed between ML and suicide rates among male youth. | Mixed |
| Hawke [56] | 2021 | Canada | Ontario | RL | Cross sectional study | Pre-RCL | 269 (pre-legalization [n = 101], post-legalization [n = 168]) | Youth Addictions and Concurrent Disorder Service (YACDS) | Internalizing disorders (e.g., depression, anxiety) Externalizing disorders (e.g., conduct disorder | No evidence of change in mental health symptomatology - internalizing disorders (e.g., depression, anxiety) and externalizing disorders (e.g., conduct disorder) - after recreational legalization. | No impact |
| Kim [57] | 2024 | Canada | Ontario | RL | Quasi-experimental study | Pre-RCL | 21,061 | National Ambulatory Care Reporting System (NACRS) | Cannabis-related ED visits combined with psychosis-related ED visits All mental health ED visits | RL was associated with reductions in cannabis-related, mental health-related, and cannabis-plus-psychosis-related ED visits among schizophrenia patients. | Positive |
| L'Heureux [58] | 2023 | Canada | Quebec | RL | Cohort study (Retrospective) | Pre-RCL | 2,448 | Hospital documents from three psychiatric EDs in Quebec City | ED consultation for a psychotic episode | No significant evidence of change in ED consultation for a psychotic episode after recreational legalization. | No impact |

*(Continued)*

| Lead author | Year of publication | Country | Province(s)/state(s) | Policy framework | Study design | Comparison/control group | Sample size | Data source | Outcome | Findings | Overall impact[a] |
|---|---|---|---|---|---|---|---|---|---|---|---|
| Lira [59] | 2024 | United States | Nationwide | ML RL COM | Cross sectional study | States which did not adopt drug policy | 272,897 | National Violent Death Reporting System (NVDRS) Restricted Access Database | Suicide Deaths of undetermined intent (UI) | More liberal state-level cannabis policies were linked to higher odds of cannabis involvement in suicides and unintentional injury deaths. | Negative |
| Marinello [60] | 2023 | United States | Colorado Washington Oregon Alaska Nevada California Massachusetts | RL | Quasi-experimental study | Pre-RCL | 93,870 | Detailed Multiple Cause of Death Research Files from 2009 to 2019, obtained from the National Center for Health Statistics' Division of Vital Statistics | Suicide rate | No significant evidence of change in suicide rates after recreational legalization. | No impact |
| Matthay [61] | 2021 | United States | Nationwide | ML RL | Cohort study | States which did not adopt drug policy Pre-RCL Pre-MCL | 75,395,344 | Claims data on commercial and Medicare Advantage health plan beneficiaries | Self-harm injuries, including intentional non-suicidal self-harm (e.g., cutting) and suicide attempts (e.g., intentional drug overdose) | No evidence of association between medical legalization and self-harm injuries. Relative increase (non-significant) in rates of self-harm injuries in states with recreational legalization. | No impact |
| McCarthy [62] | 2024 | Canada | Ontario | ML RL COM | Cross sectional study | Pre-RCL Pre-MCL Pre-COM | 439,700 | Ontario Health Insurance Plan (OHIP) data | ED visits for anxiety disorders with cannabis involvement | The rate of ED visits for anxiety disorders involving cannabis increased by 156% during the cannabis COM/COVID-19 period compared to the pre-legalization period. | Negative |

*(Continued)*

**Table 1.** (Continued)

| Lead author | Year of publication | Country | Province(s)/ state(s) | Policy framework | Study design | Comparison/ control group | Sample size | Data source | Outcome | Findings | Overall impact[a] |
|---|---|---|---|---|---|---|---|---|---|---|---|
| Mennis [63] | 2023 | United States | Nation-wide | RL | Quasi-experimental study | States which did not adopt drug policy Pre-RCL | 600 state-year observations | National Survey on Drug Use and Health (NSDUH) interviews | Depression (Major depressive episode) | Higher mean prevalence of depression and cannabis use in states that enacted recreational legalization. Stronger association between depression and cannabis use after recreational legalization of cannabis. | Negative |
| Moran [64] | 2022 | United States | Nation-wide | ML RL | Cross sectional study | States which did not adopt drug policy | 25,814 | 2017 National Inpatient Sample (NIS) Healthcare Cost and Utilization Project (HCUP) | Proportion of hospitalizations for psychosis associated with cannabis use out of all hospital discharges | A higher proportion of hospital discharges for psychosis related to cannabis use in liberal cannabis legalization policies in the United States. | Negative |
| Myran [65] | 2023/a | Canada | Ontario | RL COM | Cohort study | Pre-RCL Pre-COM | 105,203 | ICES (formerly known as the Institute for Clinical Evaluative Sciences) | Cannabis-induced psychosis | No change in rate of cannabis-induced psychosis after strict recreational legalization, while significantly higher ED visits with cannabis-induced psychosis following the commercialization. | Mixed |
| Myran [66] | 2023/b | Canada | Ontario Quebec Alberta British Columbia | RL COM | Cross sectional study | Pre-RCL Pre-COM | 5,374 | Discharge Abstract Database (DAD) and the Hospital Morbidity Database | Cannabis-induced psychosis | Significant increase in hospitalization rates for cannabis-induced psychosis after commercialization of cannabis. | Negative |
| Myran [67] | 2024 | Canada | Ontario | ML RL COM | Cross sectional study | Pre-RCL Pre-COM | 158,912 | De-identified and linked health administrative databases housed at ICES | ED visits for self-harm with cannabis involvement | Cannabis-related self-harm injury ED visits rose significantly following ML but showed no increase during the RL or the later COM/COVID-19 period. | Mixed |

*(Continued)*

| Lead author | Year of publication | Country | Province(s)/state(s) | Policy framework | Study design | Comparison/control group | Sample size | Data source | Outcome | Findings | Overall impact[a] |
|---|---|---|---|---|---|---|---|---|---|---|---|
| Ortega [68] | 2023 | United States | Nationwide | RL | An event-study within a difference-in-differences framework | States which did not adopt drug policy Pre-RCL | 647 state-year observations | SAMHSA's Uniform Reporting System (URS) at state-level | Mental health treatment admissions | Significant decrease (37%) in total mental health admissions in the states with recreational marijuana legalization. | Positive |
| Reece [69] | 2020 | United States | Nationwide | RL ML DCR | Ecological Cohort study | States which did not adopt drug policy | 410,138 | NSDUH SAM-HSA substate shape-files for 2010–2012 and 2014–2016 | Serious mental illness Any mental illness Major depressive illness Suicidal ideation | Little evidence (non-significant) of increase in SMI, any mental illnesses and suicidal ideation after decriminalization and legalization. | No impact |
| Rich [70] | 2020 | United States | Nationwide | ML RL | Quasi-experimental study | States which did not adopt drug policy Pre-RCL Pre-MCL | 67,500 | National Center for Health Statistics and National Survey on Drug Use and Health (NSDUH) | State-level suicide mortality rate Mental health morbidity | A significant reduction in suicide rates for males after medical legalization. No significant evidence of change in overall suicide rates or other mental health comorbidities after medical legalization. | Mixed |
| Rylander [71] | 2014 | United States | Colorado | ML | Cohort study (Retrospective) | Pre-MCL | N/A | The Colorado Department of Public Health and Environment (CDPHE) registry of medical marijuana users and database of completed suicides | Completed suicides | No significant evidence of association between the number of medical marijuana registrants and completed suicide after medical legalization of cannabis in Colorado. | No impact |
| Vignault [72] | 2021 | Canada | Quebec | RL | Cohort study (Retrospective) | Pre-RCL | 1,247 | Digitized hospital records from two CHUS hospitals | Diagnosis of psychotic disorders Personality disorder | No significant evidence of change in the psychotic disorder diagnoses after recreational legalization. A significant increase in the proportion of patients with a personality disorder after recreational legalization. | Mixed |

*(Continued)*

Table 1. (Continued)

| Lead author | Year of publication | Country | Province(s)/ state(s) | Policy frame-work | Study design | Comparison/ control group | Sample size | Data source | Outcome | Findings | Overall impact[a] |
|---|---|---|---|---|---|---|---|---|---|---|---|
| Wang [73] | 2022 | United States | Colorado | RL COM | Cohort study | Counties which did not have dispensaries Pre-RCL | Schizophrenia [n = 28,623]; Psychosis ED visits [n = 56,967] | Colorado Hospital Association (CHA) hospital discharge data | Psychosis Schizophrenia | Increase (24%) in the rate of psychosis visits in Colorado after recreational legalization and commercialization of cannabis, regardless of prior exposure to medical dispensaries. No significant evidence of change in the rate of schizophrenia following the recreational legalization. | Mixed |
| Wiese [74] | 2024 | Canada | Ontario | RL | Qualitative study | None | 20 | Semi-structured interviews | Impacts on patients' mental health | Legalization has increased access to high-THC cannabis, worsening psychosis, anxiety, especially in individuals with psychotic disorders, as seen in more ED visits and consultations. | Negative |
| Yeung [75] | 2020 | Canada | Alberta | RL | Quasi-experimental study | Pre-RCL | 4,732 | National Ambulatory Care Reporting System (NACRS) Alberta poison control HealthLink (a public telehealth service covering all of Alberta) | Non-mood psychotic disorder Mood disorder Anxiety-related disorder Adult personality and behavioural disorder | Significant decreases in psychotic disorder (-21%), mood-related disorder (-30%), personality and adult behavioural disorder (-25%) and anxiety disorders (-14%) after recreational legalization. | Positive |

*(Continued)*

| Lead author | Year of publication | Country | Province(s)/ state(s) | Policy framework | Study design | Comparison/ control group | Sample size | Data source | Outcome | Findings | Overall impact[a] |
|---|---|---|---|---|---|---|---|---|---|---|---|
| Zellers [76] | 2023 | United States | Nationwide | RL | Quasi-experimental study | Co-twins living in states which did not adopt drug policy | 4,043 | Longitudinal community twin samples maintained by the Minnesota Center for Twin Family Research and the Colorado Center for Antisocial Drug Dependence | Negative affect Psychoticism Externalizing behavior Detachment | No significant evidence of association between psychoticism and recreational legalization | No impact |

ML: Medical Legalization, RL: Recreational Legalization, COM: Commercialization, DCR: Decriminalization, ED: Emergency Department, SMI: Serious Mental Illness, PTSD: Post-Traumatic Stress Disorder, N/A: Not available

[a]Overall policy impact on mental health outcomes. A positive impact indicates improvement in mental health symptoms or a reduction in adverse outcomes. A negative impact indicates worsening symptoms or an increase in adverse outcomes. [b]No significant impact of the policy on mental health outcomes (i.e., null or inconclusive findings). [c]Mixed results were observed across different subgroups or outcome measures.

**Table 2. Summary of Review Papers.**

| Lead Author | Year of Publication | Country | Province(s)/state(s) | Policy Framework | Study Design | Number of Included Studies | Overlapping Original Papers[a] | Outcome | Findings | Overall Impact[b] |
|---|---|---|---|---|---|---|---|---|---|---|
| Bahji [77] | 2019 | Globally | N/A | ML RL | Systematic review | 36 | [49,54] | SMI Cannabis-related ED visits | The prevalence of SMI is significantly higher after medical legalization in states that enacted the law. Notable increase in the number of cannabis-related ED visits with psychiatric comorbidities following recreational legalization. Significant controversy about the overall impact of legalization on mental health. | Negative |
| Boury [78] | 2022 | Canada | Nationwide | RL | Narrative review | N/A | [45] | Cannabis-related hospitalizations/morbidity | Mixed results from studies in Canada on the impact of legalization on cannabis related-hospitalizations. Some studies reported increase in ED presentation for mental health problems related to cannabis while the other ones did not; and the increases were mostly non-significant comparison before legalization and post-legalization periods. | Mixed[c] |
| Cantor [79] | 2024 | Globally | N/A | COM | Systematic review | 32 | [50,61,65,73] | Self-harm Cannabis-induced psychosis Psychosis-related diagnoses ED visits for psychosis or schizophrenia | No evidence indicating a significant link between increased cannabis retail access and health outcomes potentially related to cannabis, such as self-harm, or psychosis. | No impact[d] |
| Chiu [80] | 2021 | United States | Nationwide | ML RL | Narrative review | N/A | None | Cannabis-related health service presentations | There have been reports of increase in cannabis-related presentations to health services - including psychiatric presentations - after the legalization. More studies with better designs and longer time coverage needed for better assessment of legalization impact on health outcomes. | Negative |

(Continued)

Table 2. (Continued)

| Lead Author | Year of Publication | Country | Province(s)/ state(s) | Policy Framework | Study Design | Number of Included Studies | Overlapping Original Papers[a] | Outcome | Findings | Overall Impact[b] |
|---|---|---|---|---|---|---|---|---|---|---|
| Crocker [81] | 2021 | Globally | N/A | ML RL | Narrative review | N/A | [41,45] | Cannabis ED representations for mental health concern | There is limited literature on ED presentations with cannabis-related mental health concerns comparing pre-post legalization. Existing studies in legalized jurisdictions, postulate an overall increase in mental health related adverse effects of cannabis following legalization. However, there appears to be increases in cannabis-related ED visits in jurisdictions that have not enacted legalization, possibly as a result of change of attitude towards cannabis in societies. | Negative |
| Farrelly [82] | 2023 | Globally | N/A | RL | Systematic review | 65 (6 studies on mental health outcomes) | [51,52,56,64,72] | Cannabis-related Emergency Service Utilization Psychotic disorders Hospital admissions with schizophrenia Co-occurring personality and mood-related co-diagnoses Anxious mood fluctuations in adolescents Mental health symptomology | The evidence on cannabis-related mental outcomes is mostly mixed, or they found no association between recreational legalization and mental health outcomes. | Mixed |
| Fortier [29] | 2024 | Globally | N/A | RL | Scoping review | 28 | [41,45,48,50,51,56,61, 64,72,73,75,76] | Psychosis Suicide and suicidality Depression and mood disorder Anxiety Other mental issues | Limited research on RCL shows mixed results, with increased cannabis-induced psychosis in ED visits and potential negative effects on mood and anxiety, but no impact on overall suicide rates despite higher cannabis toxicology findings. | Mixed |
| Hall [83] | 2023 | Canada | Nation-wide | RL | Systematic review | 19 (5 on mental health related outcomes) | [41,45,72] | Adult psychiatric presentations (cannabis-induced psychosis and mental health presentations) | There is mixed evidence of increases in psychosis and mental health ED presentations after legalization, however non-significant or unreliable. | Mixed |

*(Continued)*

| Lead Author | Year of Publication | Country | Province(s)/state(s) | Policy Framework | Study Design | Number of Included Studies | Overlapping Original Papers[a] | Outcome | Findings | Overall Impact[b] |
|---|---|---|---|---|---|---|---|---|---|---|
| Hammond [84] | 2023 | United States | Nationwide | ML RL | Narrative review | N/A | [38,49] | SMI Suicide-related outcomes | Findings from one study suggest that liberal policy is linked to a higher prevalence of SMI, with cannabis use contributing in part to this connection. Overall, data on the effects of RCL and MCL on SMI, suicide-related outcomes, and mortality remain limited, and present conflicting results. | Mixed |
| Hinckley [85] | 2022 | United States | Colorado | RL | Narrative review | N/A | [54] | Mental health diagnoses in ED visits Suicide | Observed increases in prevalence of cannabis-related ED visits with mental health diagnoses, and rates of suicide with cannabis present in toxicology. | Negative |
| Lake [86] | 2019 | Canada | Nationwide | RL | Narrative review | N/A | [38,47,53,71] | Incidence and prevalence of psychosis and psychotic disorders (e.g., schizophrenia) overall and across risk demographics (e.g., youth Prevalence of depression and anxiety overall and across risk demographics Incidence of attempted and completed suicide overall and across risk demographics (e.g., young adult males) | No studies were found on the impact of cannabis policy of rates of schizophrenia, psychosis, anxiety, depression. No significant association between medical legalization and suicides by longitudinal studies. | No evidence No impact |
| Leung [87] | 2019 | United States | Nationwide | ML RL | Narrative review | N/A | [38,53,54,71] | Schizophrenia and other psychotic disorders Suicide and intentional self-harm Depression | No studies were identified on the impact of medical or recreational legalization on psychosis and depression. No significant evidence on the association between cannabis policies and suicide rates. | No evidence No impact |
| Manthey [88] | 2023 | Globally | N/A | ML RL COM | Systematic review | 13 | [73] | Psychosis and schizophrenia | Increase in cannabis-related health outcomes including psychosis following legalization and opening of legal retailers. | Negative |

*(Continued)*

**Table 2.** (Continued)

| Lead Author | Year of Publication | Country | Province(s)/state(s) | Policy Framework | Study Design | Number of Included Studies | Overlapping Original Papers[a] | Outcome | Findings | Overall Impact[b] |
|---|---|---|---|---|---|---|---|---|---|---|
| Rubin-Kahana [89] | 2022 | Canada | Nationwide | RL | Narrative review | N/A | [41,56,72] | Cannabis-related hospitalizations Psychotic disorder | No significant change in the cannabis-related hospitalizations and diagnoses of psychotic disorder among youth, comparing pre- and post- legalization. | No impact |
| Walker [90] | 2023 | Globally | N/A | RL COM | Systematic review | 29 | [41,45,48,50,61,73,75] | Cannabis-induced psychosis/ schizophrenia/ psychosis self-harm and death by suicide | The evidence around recreational legalization and commercialization and mental health outcomes is limited, low-quality and mixed. | Mixed |
| Zvonarev [91] | 2019 | United States | Nationwide | ML RL | Narrative review | 47 | None | Suicide | Increase in the rate of suicide in several states after legalization. | Negative |

ML: Medical Legalization, RL: Recreational Legalization, COM: Commercialization, DCR: Decriminalization, ED: Emergency Department, SMI: Serious Mental Illness, PTSD: Post-Traumatic Stress Disorder N/A: Not available

[a]Original studies included in this scoping review that were also reviewed in the included review papers. [b]Overall policy impact on mental health outcomes. A positive impact indicates improvement in mental health symptoms or a reduction in adverse outcomes. A negative impact indicates worsening symptoms or an increase in adverse outcomes. [c]Mixed results were observed across different subgroups or outcome measures. [d]No significant impact of the policy on mental health outcomes (i.e., null or inconclusive findings).

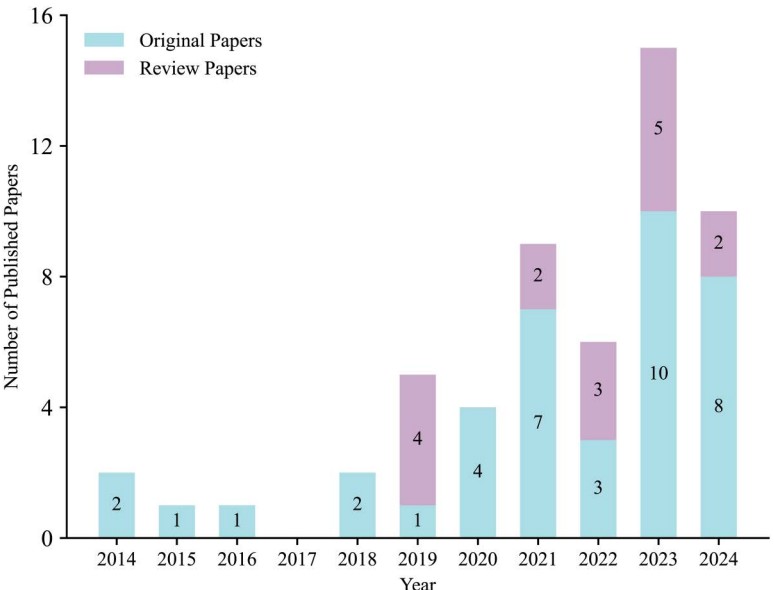

**Fig 4. Number of included original and review papers by year of publication.**

Most of the review papers examined a broader range of outcomes beyond mental health conditions. Drug use patterns, overdose rates, SUD prevalence, cannabis-related traffic incidents and physical health issues were among the outcomes analyzed, alongside non-SUD mental health outcomes. Among these, psychosis was the most commonly explored non-SUD mental health outcome [29,79,82,83,86–90], followed by, suicide or self-harm [29,79,84–87,90,91], psychiatric ED visit or hospitalization [77,78,80–83,85,89], depression [29,86,87], anxiety [29,82,86], and SMI [84,94] (See Fig 3).

**Overall policy impact on mental health outcomes**

Given the substantial variation in policy frameworks, study designs, and reported outcomes, this section does not attempt to quantify the strength of evidence or draw conclusions tied to specific legal contexts. Instead, we offer a high-level, descriptive summary of the overall direction of findings related to mental health outcomes. The effects of drug policy reforms can differ significantly across jurisdictions, shaped by local enforcement practices, health system capacity, and supporting regulations. Therefore, the findings should be interpreted with caution and understood as indicative of general trends rather than direct comparisons between studies conducted in diverse settings.

Fig 5 illustrates the distribution of the "overall impact" of implemented drug policies on mental health outcomes in both original studies and review articles. A notable majority of original studies found no significant evidence linking policy enactment to changes in mental health outcomes (n = 14) [41,45–47,50,53,56,58,60,61,69,71,76], a conclusion also supported by four review papers [86,87,89]. One-quarter of original studies [49,51,54,63,64,66] and over one-third of review papers [80,81,85,88,91,94] reported increases in the prevalence, incidence, or trend of adverse mental health outcomes, such as psychosis and depression, suggesting a potentially negative impact. Conversely, six original studies (15%) reported a positive impact, noting reductions in adverse mental health outcomes [38,39,42,43,68,75]; however, no review papers corroborated this finding. Lastly, over 20 percent of original studies [48,52,65,70,72,73] and six of sixteen reviews [78,82,83,90] presented mixed findings, pointing to inconclusive nature of the existing evidence.

Studies from both the United States and Canada predominantly found that cannabis policies had no significant impact on mental health outcomes, with 34.5% of U.S. studies and 38.9% of Canadian studies reporting no overall effect. The

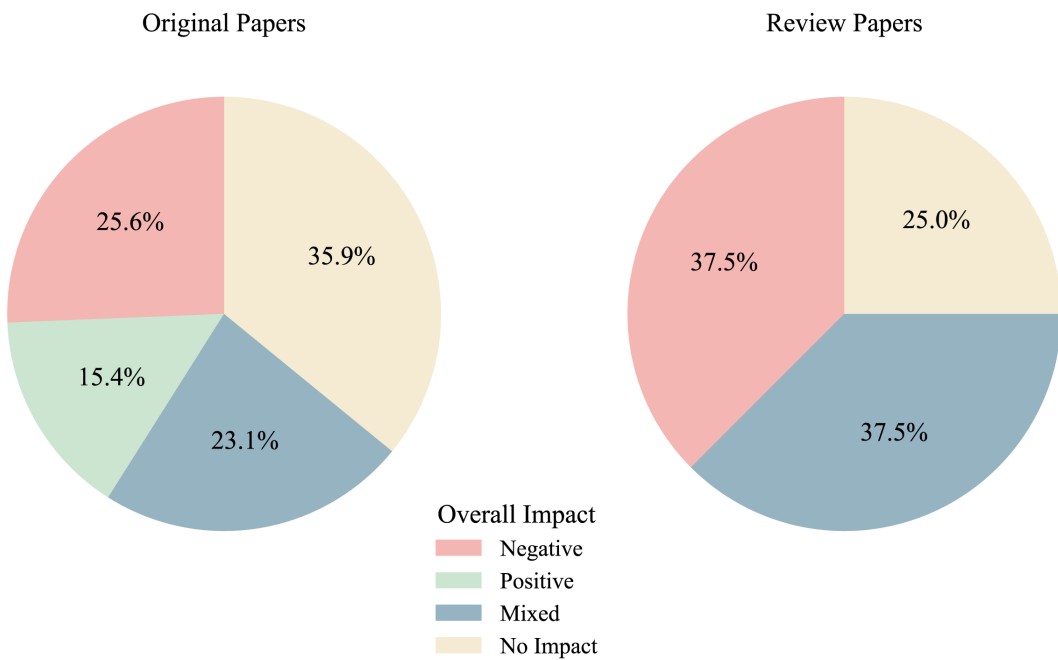

Original Papers Review Papers

**Overall Impact**
- Negative
- Positive
- Mixed
- No Impact

**Fig 5. Overall policy impact on mental health outcomes in original papers versus review papers.**

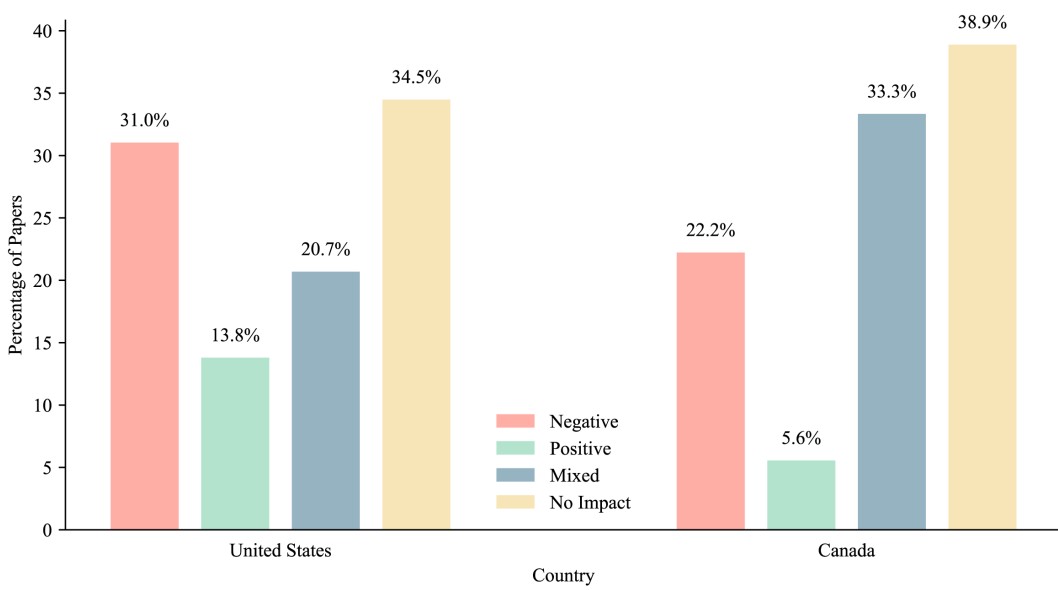

**Fig 6. Overall policy impact on mental health outcomes in the United States versus Canada.**

second most common finding in U.S. studies was a negative overall impact (31%), whereas in Canadian studies the second most frequent was mixed finding (33.3%). A comparison of overall policy impacts between studies from the U.S. and Canada is presented in Fig 6.

## Discussion

This scoping review aimed to provide a landscape of the existing evidence on the impact of liberal drug legislation frameworks on non-SUD mental health outcomes and the prevalence of co-occurring mental conditions. The existing literature on the mental health impacts of liberal drug policies is both limited and inconsistent. Findings are often shaped by local contexts, which makes it difficult to draw generalizable conclusions across jurisdictions. A substantial gap exists regarding the implications of "decriminalization of non-cannabis psychoactive drugs "on mental health outcomes. Notably, despite our primary interest in exploring the impact of liberal policies targeting drugs other than cannabis, our search did not yield any published studies or policy documents on the mental health impacts of non-cannabis laws.

One potential explanation is that in jurisdictions prioritizing decriminalization, the primary goal of policy shift is to address critical outcomes, such as overdose and the incidence of HIV/AIDS [5,95]. In these settings, impact on mental health and co-occurring mental disorders might seem less urgent and are consequently overlooked in policy evaluations. Similarly, for cannabis policies, mental health outcomes may be perceived as indirect and long-term, making them less timely to investigate. Thus, they are typically studied alongside more immediate and short-term outcomes, such as changes in pattern of use, intoxication, and the incidence of cannabis use disorders.

The existing literature is further complicated by potential biases due to differing perspectives on the impacts of liberal drug policies. As discussed in the introduction, two major perspectives exist regarding the implications of such policies. Supporters of liberal drug regulations view these changes as a promising avenue to reduce criminal charges against drug users and redirect them to health and treatment services. This approach, they argue, will reduce the overdose deaths and, in turn, alleviate the social, mental, and physical burdens on individuals, the government, and the justice system [96].

Conversely, some assert that non-punitive strategies along with increased drug availability might encourage first-time use and lead to chronic and problematic use. Cannabis is often cited by this group as a "gateway drug," based on the perception that its relative safety could lead individuals to experiment with more harmful substances such as opioids and stimulants [6,80]. However, it is important to note that there is no conclusive evidence supporting a causal relationship between cannabis use and the use of more dangerous substances [97]. We interpret the reviewed papers as reflecting differing viewpoints within existing research. These potential biases may have contributed to the mixed and sometimes contradictory findings, making it challenging to draw definitive conclusions.

Most review papers around the effects of drug policies primarily focus on public health implications, often highlighting their potential negative impacts [98]. While mental health is considered an indicator of population well-being, it is typically evaluated alongside other metrics such as drug use rates, overdose incidents, traffic accidents, and physical morbidities like cannabis-induced hyperemesis. Our findings regarding the most frequently studied mental health outcomes are congruent with those reported by Fortier et al, with psychosis, suicide, and depression consistently identified as the most examined conditions in the context of liberal drug policies [29]. While these three conditions have been extensively studied, there remains a need for research into their long-term patterns and trajectories of these conditions. Additionally, greater attention should be directed toward other mental health conditions underrepresented in the current body of research, including anxiety, mood disorders, personality disorders, and sleep disturbances.

Moreover, our findings indicate that the majority of research in this field emerged after the U.S. legalized cannabis for recreational use. The earliest study, published in 2014, examined the effects of Colorado's medical cannabis legalization on suicide rates [71].This is notable because many jurisdictions had already implemented liberal drug policies, especially cannabis decriminalization and medical legalization, by the late 1990s and early 2000s [92,93]. While some studies retrospectively analyzed data from these earlier periods [38,39,42,47,53], they were not published until the late 2010s. This delay may indicate that the mental health implications of drug policies were not initially prioritized, and became a focus only after concerns arose about potential links between heavy cannabis use and mental health conditions, such as psychosis and suicide [99,100]. Since then, this link has been subjected to numerous investigations [101]. However, as this

review has highlighted, only a limited number of studies have examined the role of drug regulations when analyzing the connection between cannabis use and non-SUD mental disorders.

Another key finding is the lack of evidence from countries outside North America. This observation aligns with a global mapping review by [102] in which 427 out of 438 included studies were conducted in North America. This finding was similar to a previous scoping review by Fortier that reported original studies exclusively from Canada and the United States, with no articles originating from other countries with cannabis legalization [29]. This imbalance raises questions about the generalizability of findings to regions with different socio-political contexts, such as Europe and South America. Although Europe has a long history of liberal drug laws [103], particularly in the Netherlands, Portugal, Spain, and the Czech Republic, our review did not identify studies assessing the mental health impacts of drug legislation in these areas. Nevertheless, it is important to acknowledge that this gap may stem from the exclusion of non-English literature and non-publicly available documents in our scoping review.

Comparing the mental-health impact of liberal drug policies across jurisdictions is inherently challenging because outcomes depend not only on the policy text but also on how—and how well—the policy is implemented. Portugal's landmark 2001 reform, for instance, paired nationwide decriminalization with substantial investment in treatment and harm-reduction programs through its "Commissions for the Dissuasion of Drug Addiction," a package linked to declines in drug-related harms and psychiatric sequelae [104]. British Columbia has opted for a slower, pilot-based approach within a tighter legislative framework [105,106], while Oregon's 2021 decriminalization earmarked additional health funding that some observers view as inadequate, potentially blunting mental-health benefits [107].

These examples underscore that drug policies are not uniform interventions: local enforcement practices, health-system capacity, and complementary regulations can strongly influence psychiatric outcomes. It is also important to consider that these outcomes may be influenced by variability in supply safety and regulatory frameworks, such as differences in product potency, quality control, and access. Weak or absent regulations often result in untested, high-potency substances lacking quality standards or consumer information. For example, in the UK, street cannabis often contains high THC with little CBD, a profile linked to heightened risk of psychosis [108]. Likewise, in British Columbia, Canada, drug checking data show that the illicit opioid supply is highly variable, with frequent contamination involving fentanyl. About half of analyzed opioid samples also contained benzodiazepines [109]. The polysubstance nature of the unregulated supply, combined with the high variability of the compounds, increases the risk of overdose and hospital admissions and underscores how unregulated drug markets produce unpredictable, high-potency substances with serious public health consequences [110].

We acknowledge that the exclusion of non-English literature may have led to an underestimation of evidence from certain countries, particularly in Europe and South America, and excludes localized articles and policy reports published in non-English media that may offer important context in regions where decriminalization policies have been implemented. Furthermore, our review did not extract or analyze contextual factors, concurrent interventions, or specific regulatory elements associated with the policies examined. Lastly, our review did not involve consultation with individuals with lived experience, third-sector or governmental organizations, or experts by experience. Engaging these groups in future research could help include diverse perspectives and improve the scrutiny of drug policy impacts in relation to specific settings, conditions, and populations.

Despite its limitations, this review offers a broader perspective than Fortier et al. [29] by synthesizing findings from both original and review papers to locate all relevant publications. With extended scope and timeframe our review provides a more comprehensive overview of the current evidence, and highlights significant gaps in knowledge regarding study design, setting, and time. These findings underscore the critical need for more comprehensive and methodologically rigorous research to deepen our understanding of the mental health impacts of less punitive drug policies.

To generate evidence that is both generalizable and policy-relevant, future studies must move beyond basic outcome assessments and adopt designs that account for broader policy and social contexts. This includes pre-policy conditions,

the scale of harm reduction and public health efforts, implementation funding, and the availability of mental health and addiction services. A key consideration is that study designs should address temporality by incorporating longer follow-up periods, as well as by capturing the sequencing of events, allowing for causal inferences. Current approaches often demonstrate associations without clarifying whether mental health outcomes follow substance use or policy changes. Causal inference would be strengthened if the timing of substance use initiation were clearly measured relative to the onset of mental health conditions.

Importantly, future research should also address the clear gap in evaluations of policy reforms targeting substances other than cannabis. Qualitative and mixed-methods approaches that incorporate the lived experiences of people who use drugs, community members, service providers, and policymakers, can reveal unintended consequences, community dynamics, and contextual factors shaping policy impact. By integrating these perspectives, research can better explain cross-jurisdictional variation and support the development of more nuanced, equitable, and effective drug policies.

## Conclusions

This review underscores a significant evidence gap in understanding the mental health impacts of liberal drug policies— particularly those targeting substances beyond cannabis. While research has increasingly addressed outcomes such as substance use disorders and overdose, non-SUD mental health conditions remain underexplored and inconsistently reported. The limited and context-dependent nature of existing findings highlights the need for more rigorous, long-term studies that account for policy design, implementation setting, and population differences. Addressing these gaps is essential to support evidence-informed, equitable drug policy reforms that consider both intended and unintended mental health outcomes.

## Supporting information

**S1 Appendix.  Example of search strategy ran in EMBASE.**
(DOCX)

**S2 Appendix.  Data extraction tool.**
(DOCX)

**S3 Appendix.  PRISMA-ScR checklist.**
(PDF)

## Acknowledgments

We are grateful to Ursula Ellis for her guidance in developing the search strategy for this scoping review. We would also like to thank Hamidreza Aftabi for his assistance in generating the plots using Python, which enhanced the visual representation of this work.

## Author contributions

**Conceptualization:** Mana Mohebbian, Joseph Puyat.

**Data curation:** Mana Mohebbian, Sara Najafi, You Na Choi.

**Investigation:** Mana Mohebbian, Sara Najafi, You Na Choi.

**Methodology:** Mana Mohebbian, Christian Schütz, Rosemin Kassam, Arminee Kazanjian, Joseph Puyat.

**Project administration:** Mana Mohebbian.

**Supervision:** Joseph Puyat.

**Visualization:** Mana Mohebbian.

**Writing – original draft:** Mana Mohebbian.

**Writing – review & editing:** Christian Schütz, Rosemin Kassam, Arminee Kazanjian, Joseph Puyat.

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
