## [Decision Letter · Decision Letter 0]

24 Mar 2025

PMEN-D-25-00024

Exploring the impact of drug decriminalization and legalization policies on mental health outcomes: A scoping review

PLOS Mental Health

Dear Dr. Mohebbian,

Thank you for submitting your manuscript to PLOS Mental Health. After careful consideration, we feel that it has merit but does not fully meet PLOS Mental Health’s publication criteria as it currently stands. Therefore, we invite you to submit a revised version of the manuscript that addresses the points raised during the review process.

We look forward to receiving your revised manuscript.

Kind regards,

Craig Nicholas Cumming

Academic Editor

PLOS Mental Health

Journal Requirements:

1. Figure 2: please (a) provide a direct link to the base layer of the map (i.e., the country or region border shape) and ensure this is also included in the figure legend; and (b) provide a link to the terms of use / license information for the base layer image or shapefile. We cannot publish proprietary or copyrighted maps (e.g. Google Maps, Mapquest) and the terms of use for your map base layer must be compatible with our CC-BY 4.0 license.

Additional Editor Comments (if provided):

Please carefully review the important points made by the two expert reviewers. In particular attention should be paid to their comments around the language used, to ensure that you describe substances and substance use in a way that is contemporary and non-stigmatising. Also, the bi-directionality of the relationship between substance use and mental health issues should be acknowledged in greater detail to reflect the sizeable evidence base showing that causation can flow in either direction.

In your analysis and discussion around the “direction of evidence” starting on p24, there appears to be some important context missing. One of the issues is that while different jurisdictions may implement policies aimed at decriminalizing or legalizing the use of substances that were previously illegal to use, the circumstances in which they do this can differ greatly. For example, in Portugal, alongside decriminalisation of the use of previously illicit substances, there was also a substantial increase in investment in health interventions so that people identified as having a problem with substance use could be referred into treatment, and away from the justice system. The evidence suggests that the investment in health resources was largely sufficient to meet demand, with notable decreases in the rates of drug-related issues such as mental illness, overdoses and blood borne viruses as a result. This contrasts with the experience in Oregan which also decriminalized formerly illicit substance use. While additional funding was allocated to health services to help treat substance-related problems, the level of investment in these services has largely been viewed as inadequate to meet demand. Considering these contextual factors is important when interpreting the literature from different jurisdictions and assessing whether a change in drug policy also included sufficient investment in health resources to handle increased demand created by diverting people away from the justice system, and into health services. This will have an important impact on whether the rates of substance-related mental health issues changed, and in what direction they moved. Not giving consideration to contextual factors such as these will limit how well your review informs future policy and practice in the area.

Reviewers' comments:

Reviewer's Responses to Questions

**Comments to the Author**

1. Does this manuscript meet PLOS Mental Health’s publication criteria ? Is the manuscript technically sound, and do the data support the conclusions? The manuscript must describe methodologically and ethically rigorous research with conclusions that are appropriately drawn based on the data presented.

Reviewer #1: Partly

Reviewer #2: Partly

2. Has the statistical analysis been performed appropriately and rigorously?

Reviewer #1: N/A

Reviewer #2: N/A

3. Have the authors made all data underlying the findings in their manuscript fully available (please refer to the Data Availability Statement at the start of the manuscript PDF file)?

Reviewer #1: Yes

Reviewer #2: Yes

4. Is the manuscript presented in an intelligible fashion and written in standard English?

Reviewer #1: Yes

Reviewer #2: No

5. Review Comments to the Author

Reviewer #1: This paper reports on a scoping review conducted to explore any literature about the impact of drug policies (namely decriminalisation, legalisation, and commercialisation) on mental health outcomes. The paper specifically seeks to expand previous reviews of cannabis policies and mental health outcomes to include drugs other than cannabis. This paper also includes an extended timeframe and broader range of cannabis policies.

A huge strength of this paper is the reporting and synthesising of both reviews and original papers. This helps to interpret the findings in relation to previous reviews, identify areas with less clarity, and areas where the evidence base is stronger. It is also an incredibly useful finding that there does not appear to be any research on drugs other than cannabis in relation to MH outcomes.

My main feedback for this paper is that it could be strengthened by focusing less on the direction of evidence and more on the study designs, policy frameworks, and outcomes examined. Given this is a scoping review, I feel at times that the direction of evidence requires further contextualising, which is difficult due to the wide range of methods, outcomes, and policy interventions studied. At present, I believe some of the studies are being misrepresented as statements about direction of evidence are being made. An example is Lira (45) – stricter cannabis policies associated with less cannabis involvement in suicides/other deaths is summarised as ‘increase’, but I’m unclear about the assumption here (as it does not state whether suicides increased or decreased, only whether cannabis was involved). It may be that more people use cannabis, but not whether the increase in use is also associated with increases in suicide.

I’ve included a short description from another paper about scoping reviews which may assist:

“Scoping reviews synthesize evidence and assess the scope of literature on a topic. The value of adopting a scoping review methodology lies in its broader approach to mapping literature and addressing research questions than other systematic reviews. Unlike systematic reviews, the aim of a scoping review is to map the key concepts that underpin a research area (Arksey & O’Malley, Citation2005). Scoping reviews are commonly conducted to summarize and disseminate research findings, identify research gaps, and make recommendations for future research. But they may also be carried out to determine the way the research has been conducted (Peters et al., Citation2015); that is, to provide useful insight into the nature of a key research concept, to clarify working definitions, and identify conceptual boundaries of a topic or field (Peters et al., Citation2015, Citation2021).”

https://doi.org/10.1080/26895269.2024.2447785

My suggestion is to revise to identify areas for further research and more closely examine the methods used in existing research. This might help inform evaluation of the direction of the evidence. I have some more detailed feedback which I’ve left below.

-The section about relationship between substance use and mental health (introduction, para 4) cites a lot of research about substance use disorders and mental health outcomes. There might be a conflation here about substance use and substance use disorders. Many people use substances (including legal and illegal) and do not develop substance use disorders. The relationship between substance use disorders and other mental health disorders is quite different to the relationship between substance use and mental health outcomes. I would recommend clarifying this and possibly differentiating the research to be more specific. This will also avoid unnecessary stigma towards people who use drugs. Also, it may be worth noting that legal substances, such as alcohol, also are linked to MH outcomes.

-The previous review by Fortier could be better discussed, in both the introduction and discussion. Things I would be interested to know are the timeframe they used (you note yours is a longer timeframe), their findings, how many studies were included, which outcomes they focused on, and which policy interventions. Also, in discussion you could note how many of your papers were covered in the Fortier paper, and how your findings compare/contrast.

-Line 138 – ‘occurrence of secondary mental health disorders, such as depression and anxiety, that are induced or triggered by drug use’. This might benefit from further clarification. Does this mean only MH disorders which are triggered by drug use? Are any excluded? And does secondary mean, in addition to substance use disorders? Throughout, I think this paper could benefit from greater differentiation between substance use and substance use disorders.

-Outcome section could benefit from examples. Eg. Population prevalence of MH disorders, any particular MH disorders examined… other measures? Only population/national studies or also smaller samples?

-Policy framework – I couldn’t find a definition for this. This is a part of your analysis, and some work is involved in deciding how to code the papers. This would be valuable to include, how you decided upon these categories and a simple description. Also, did any papers look at decriminalisation/other approaches other than legalisation? This would be really interesting to note as an area missing.

-Sample sizes for each of the studies might be useful, as I found it difficult to know what size population was included. Eg Some of the qualitative papers I imagine were small sample sizes, but this isn’t clear.

-I would recommend separating the qualitative studies, as I don’t know if they are best summarised with the quantitative studies. The language around ‘increases’, relationships, and influence throughout suggests largely quantitative analyses. I think being clear about the study design would help interpretation

-Some of the study outcomes might benefit from better explication. Eg Mental health medications is influenced by many other factors, and a correlation study with legalisation is problematic, as there are many state differences which may impact this.

Some comments on language/minor edits:

-The term ‘hard drugs’ is not preferable in my research area (drug policy) as it can be stigmatising to people who use drugs. There is no clear differentiator between so called ‘soft’ and ‘hard’ drugs, and the term is not preferred by people with lived experience (people who use drugs). I would recommend using the language ‘drugs other than cannabis’, and defining this early on to exclude alcohol and tobacco, etc.

-93-94 could be revised

-Line 192 – brackets (n = ) should be following the outcome. Eg anxiety (n = 11, depression (n = 21). Appears to be an error somewhere here

-B footnote is unclear in table

-Alcohol induced psychosis is included for one paper – but alcohol is noted as excluded

-Kim (43) reported as increase, but outcomes state ‘reductions in…’

-Lines 270-272 – needs clarification, error somewhere about Canada vs US

Reviewer #2: Overall Feedback:

This paper addresses an important and timely topic, and I commend the authors for their efforts. However, the current version has substantial weaknesses in clarity, definitions, methodology, and framing. The extent of changes required prior to publication suggest that a straightforward revision would not be sufficient. Once you have addressed the comments below, with additional input from co-authors and guidance from subject experts, I would encourage you to re-submit for publication.

Overall, you need to improve your specificity and clarity in communicating; particularly with regards to:

(1) Define and maintain consistent terminology regarding mental health throughout the paper

(2) It is crucial that you specify what you are including when referring to mental health outcomes, specifically, you need to acknowledge and justify that you are not including substance use disorders and addiction as mental health outcomes. Regardless of whether you are examining this as an outcome of interest, failing to acknowledge them as mental health difficulties risks perpetuating stigma and misunderstanding

(3) The term "hard drug" is controversial and lacks a universally accepted definition, it is important that you define how you are using the term. Consider specifying which substances are being referred to or using more neutral, precise terms.

(4) The term “liberal drug policies” is used repeatedly without adequate explanation or recognition of the variation across different contexts.

Specific Comments

Abstract:

Requires significant re-wording and improved specificity.

Introduction:

Paragraph 1 lacks clarity and requires re-wording to ensure communication of key messages. Define key terms such as "liberal drug policies" to ensure readers understand the framing.

Line 44: Specify UN priorities you are alluding to.

Line 47: Add references and instead of broadly noting public debate, specify in mainstream narratives this is often people lacking in specialist knowledge including politicians as well as those with differing politically aligned stances. Be specific there is no clear black/white evidence either way and there is poor public understanding which further fuels stigma, misinformation and bias.

Paragraph beginning line 59: Consider acknowledging the contrast with countries that have strict drug policies but high usage rates (e.g., France).

Line 54 and Paragraph beginning line 59: Clarify whether higher use is presented as a negative or neutral outcome. Consider contextualising the increase similarly to other commercially available goods, where greater availability leads to higher consumption?

Paragraph beginning line 71 (and overall) - The omission of addiction and substance use disorders is significant and needs addressing. Additionally, there is no acknowledgement that mental health issues are associated with substance use - the bi-directional relationship - or the global trends in worsening mental health.

Line 85 – Ultimately the review covers the same focus as Fortier et al, and it includes the same studies, the rationale needs strengthening for what this review adds – does the paper offer any new insights?

Methods:

Consider whether the absence of consultation with experts by experience, advocates, or third-sector specialists (e.g., Canadian Mental Health Association) is a limitation worth addressing. This could enrich the paper’s perspective and also provide some much-needed guidance on language.

Publication type: The exclusion of non-English studies is a notable limitation, particularly given that many countries implementing decriminalization policies are non-English speaking. Acknowledge the study is by default excluding all localised articles and policy reports.

Intervention: As noted previously, define what is meant by "hard drugs". Additionally, how are you differentiating prescribed medications (line 146) from prescribed cannabis (which is a medicine)? or prescribed psychedelics (in the case of the legalisation of psychedelic therapies)?

Outcome: Clarify the definition of secondary mental health disorders. Are these considered secondary to substance use (which is not the same as substance use disorder and addiction), or mental health crises triggered by substance use? Are you differentiating between one-time and chronic effects?

Exclusion: Expand and justify the exclusion of studies examining substance use disorder outcomes (given that this is a mental health condition).

Results

Line 191: Distinguish between self-harm and suicide – specify, do you mean suicidal ideation, survival or deaths by suicide? Categorise appropriately.

Line 192: What is the (n=21) referring to? Re-word for clarity.

Table 1: Define policy framework categories earlier in the paper for clarity.

Time trends: Given that you only include English papers, it follows, there are no papers from non-English speaking countries, so reference to Portugal here does not make sense; it would be more relevant to refer to the dates of decriminalisation in the locations of the studies included in the analysis.

Summary of original studies: Address the disparity in outcome indicators — for example, psychiatric medication fill rates may reflect prescribing trends rather than psychiatric symptomatology, making them less comparable to direct measures of mental health outcomes like depression. Clarify comparison/control groups (e.g., administrative data before/after policy implementation) and add mention of trends based on the type of study. Comment on the inclusion of the UK despite its lack of liberal drug policies. Consider also summarising trends relative to the policy frameworks examined.

Summary of review papers: Include a narrative synthesis of the findings, similar to the original studies section. Clarify which original studies have also been included in these reviews to acknowledge repetition through secondary interpretation.

Direction of evidence (Lines 270 and 272): The phrasing is inconsistent — clarify whether Canadian studies predominantly support or contrast with US findings.

Discussion:

Line 281: Ensure consistent terminology (e.g., "concurrent", “secondary”). As noted previously ensure consistent language regarding mental health throughout.

Line 283: Expand on what is meant by "sometimes unreliable," providing concrete examples.

Line 303: Clarify that while there may be a public misperception of cannabis use being a ‘gateway’ to using more harmful substances, there is no conclusive evidence supporting this causality.

Consider commenting on the variability in supply safety and differing policy frameworks e.g. the risks associated with high-potency THC.

Lines 363: Acknowledge that this review is not global, that it is only inclusive of English-speaking countries, focusing primarily on North American policies.

Line 366: Offer specific recommendations for improving study designs, addressing gaps and expanding research - avoid vague or repetitive conclusions.

Line 369: Remove the phrase "especially in regions with a long history of such policies," this is inappropriate given that the review design inevitably excludes these countries by excluding non-English content.

Additional query: Generation of the plots is a substantial contribution, unless personal preference, Hamidreza Aftabi should be a named author.

6. PLOS authors have the option to publish the peer review history of their article (what does this mean? ). If published, this will include your full peer review and any attached files.

**Do you want your identity to be public for this peer review?** For information about this choice, including consent withdrawal, please see our Privacy Policy .

Reviewer #1: No

Reviewer #2: No

---

## [Decision Letter · Decision Letter 1]

14 Aug 2025

PMEN-D-25-00024R1

Exploring the impact of drug decriminalization and legalization policies on mental health outcomes: A scoping review

PLOS Mental Health

Dear Dr. Mohebbian,

Thank you for submitting your manuscript to PLOS Mental Health. After careful consideration, we feel that it has merit but does not fully meet PLOS Mental Health’s publication criteria as it currently stands. Therefore, we invite you to submit a revised version of the manuscript that addresses the points raised during the review process.

The manuscript has improved since the first submission, but still requires some changes, as outlined in detail by the reviewer. Please carefully consider the reviewer's feedback and respond accordingly.

We look forward to receiving your revised manuscript.

Kind regards,

Craig Nicholas Cumming

Academic Editor

PLOS Mental Health

Journal Requirements:

Reviewers' comments:

Reviewer's Responses to Questions

**Comments to the Author**

1. If the authors have adequately addressed your comments raised in a previous round of review and you feel that this manuscript is now acceptable for publication, you may indicate that here to bypass the “Comments to the Author” section, enter your conflict of interest statement in the “Confidential to Editor” section, and submit your "Accept" recommendation.

Reviewer #1: All comments have been addressed

Reviewer #2: (No Response)

2. Does this manuscript meet PLOS Mental Health’s publication criteria ? Is the manuscript technically sound, and do the data support the conclusions? The manuscript must describe methodologically and ethically rigorous research with conclusions that are appropriately drawn based on the data presented.

Reviewer #1: Yes

Reviewer #2: Partly

3. Has the statistical analysis been performed appropriately and rigorously?

Reviewer #1: I don't know

Reviewer #2: Yes

4. Have the authors made all data underlying the findings in their manuscript fully available (please refer to the Data Availability Statement at the start of the manuscript PDF file)?

Reviewer #1: Yes

Reviewer #2: Yes

5. Is the manuscript presented in an intelligible fashion and written in standard English?

Reviewer #1: Yes

Reviewer #2: Yes

6. Review Comments to the Author

Reviewer #1: Thankyou for your resubmission and the chance to review this paper again. I want to acknowledge the huge amount of work that has gone into this paper and the revisions, and I believe it is a useful and worthwhile publication. Thank you for addressing the comments/feedback in such detail, and your substantial edits.

I don't have any comments/edits to provide.

Reviewer #2: Thank you for your integration of reviewer comments, it has significantly improved the paper. However, there remains a number of clarifications required prior to publication. Several sections use language that seems to imply causality, understate key advocacy arguments, and inconsistently frame mental health in relation to substance use. Addressing these issues will improve clarity, provide a more balanced and nuanced framing, aligning with existing evidence.

Throughout:

The mental health language is not consistent throughout - you switch between ‘mental health complications’ ‘mental health disorders’ ‘mental health’ as well as outdated terms ‘mental disorders’ ‘mental conditions’ etc. Health Canada often uses "mental health conditions". Pick one term and use it consistently throughout.

Throughout the paper, I believe some of the language is misleading, inferring causality without evidence nor reflective of the included studies. For example, the phrasing “substance-related mental health outcomes” implies a causal relationship however this is problematic given the bi-directional nature of the relationship and that the included studies are not causal analyses. Consider "mental health conditions in the context of substance use".

Abstract:

The current framing implies mental health is one subset of "drug-related outcomes, when mental health is central to understanding both the causes and consequences of drug use. Consider acknowledging the complex relationship upfront e.g. "concerns have emerged about their broader health and social impacts, recognising the complex bidirectional relationship between mental health and substance use".

Introduction:

Line 44, re: UN priorities – I believe "encouraging more flexible drug policies" misrepresents the UN position document, perhaps quote the UN document directly.

Line 56-63, the summary of advocates’ arguments is incomplete, centring on economic and regulatory control arguments, however, advocates typically place greater emphasis on mortality reduction and harm minimisation. Advocates primarily argue that legalisation/decriminalisation has a public health benefit and saves lives - central advocacy arguments include: preventing overdose deaths through supervised consumption and a regulated supply, expanding harm-reduction services, shifting from criminalisation to treatment and avoiding incarceration-related trauma. Economic benefits are generally framed as secondary to the public health objective of saving lives. Accurately reflecting these priorities is important for a balanced portrayal of both perspectives.

Line 83-90, this paragraph continues to frame mental health as primarily a consequence of drug use rather than acknowledging that mental health and trauma are often primary drivers of substance use. Strengthening discussion of mental health, trauma, adverse childhood experiences as drivers would create a more balanced and evidence-based framing.

Line 97, correct wording “Harmful use of legally available substances like alcohol *use disorder* can also elevate the risks of depressive and suicide”

Line 101, clarify wording “Despite clear associations, the direction of causality remains under debate.” - the causal relationship between mental health/trauma/Adverse Childhood Experiences on increased risk of substance use and dependency is well-established in the literature.

Methods:

Line 187, “While SUDs are recognized substance-related mental conditions” correct wording “are recognised mental *health* conditions”

Line 170-198, the outcome definition requires further clarification. The phrase "may develop following exposure" implies causation, but the study designs included cannot establish this, which largely show associations or temporal correlations or qualitative perspectives. Consider something like "non-SUD mental health conditions among populations with substance exposure, recognising that these conditions may be pre-existing, co-occurring, precipitated, or influenced by substance use." This will ensure the outcome definition reflects the methodological limitations of included studies and avoids overstating causal relationships.

Results:

Line 241, remove “Surprisingly”, this would be more appropriate in the discussion.

Line 306, grouping suicidal ideation, non-fatal self-harm, and suicide deaths under 'self-harm or suicide' conflates distinct phenomena with different prevalence rates, risk factors, and policy implications - please consider providing a breakdown where data permits and please label the inclusion of suicidal ideation as this is often reported without planning, intent or action, with much wider prevalence than self-harm, survival or deaths by suicide. Conflating these outcomes risks obscuring important differences in prevalence, etiology, and prevention strategies

Line 360, “However, a greater proportion of U.S. studies reported a negative overall impact, while Canadian studies more frequently indicated mixed findings.” - remove ‘however’ and please include proportions in text for clarity.

Discussion:

Line 383-388, The wording “extreme supporters” reads as bias in framing (the same language is not used for opponents) – please ensure language is consistent and balanced.

Line 387-388, As above "social, mental, and physical burdens" misses the primary argument about saving lives/reducing mortality.

Line 394, “Our review of the papers revealed that differing viewpoints are also reflected in existing research.” - this is not described in your results, is this your subjective impression or is this the framing used by the authors of the papers? Please provide this breakdown in the results section.

Line 434-442, captures the complexity required and is a good example of the nuanced analysis the paper should apply throughout.

Line 447, this wording is potentially misleading - as it doesn't acknowledge that regulation provides potency testing, labeling, and consumer choice that allows for informed decision-making. While unregulated markets have unpredictable and untested potency levels, regulated markets enable consumers to select products appropriate to their tolerance and preferences, potentially reducing adverse effects through better information and product diversity. For example see research in UK on very high THC levels with very low CBD in street cannabis or the fentanyl crisis also exemplifies how unregulated markets trend toward extremely high-potency substances with no quality control or consumer information, leading to unpredictable and often fatal outcomes.

Line 468-474, the recommendations are much improved consider including recommendations for study designs that take into account temporality - the current designs can show associations and correlations but can't rule out that observed mental health outcomes preceded substance use or policy changes as such cannot speak to causality.

7. PLOS authors have the option to publish the peer review history of their article (what does this mean? ). If published, this will include your full peer review and any attached files.

**Do you want your identity to be public for this peer review?** For information about this choice, including consent withdrawal, please see our Privacy Policy .

Reviewer #1: No

Reviewer #2: No

---

## [Editor Report · Decision Letter 2]

18 Sep 2025

PMEN-D-25-00024R2

Exploring the impact of drug decriminalization and legalization policies on mental health outcomes: A scoping review

PLOS Mental Health

Dear Dr. Mohebbian,

Thank you for submitting your manuscript to PLOS Mental Health. After careful consideration, we feel that it has merit but does not fully meet PLOS Mental Health’s publication criteria as it currently stands. Therefore, we invite you to submit a revised version of the manuscript that addresses the points raised during the review process.

We look forward to receiving your revised manuscript.

Kind regards,

Craig Nicholas Cumming

Academic Editor

PLOS Mental Health

Journal Requirements:

Additional Editor Comments (if provided):

I am close to being able to make a decision on your manuscript. I request that you please consider the following minor changes:

p.35 line 525, benzodiazepines are distinct from opioids. Are you meaning to say that benzodiazepines are being misleadingly sold as opioids, or perhaps are being mixed with opioids that are being sold? Also, you mention “a volatile and hazardous supply”. This language is quite emotive, I suggest more plain language that explains that high potency unregulated drugs often carry high risks of poor outcomes such as overdose and death, with the relevant literature cited. This will be more informative for the reader.

p. 36 line 560, longer follow-up periods are just one component of temporality that would permit causal inference, I suggest that your recommendation also includes the suggestion that studies are designed in a way that permits causal inferences to be made with respect to the relationship between substance use and mental health conditions. This would include measuring the timing of initiation of substance use and commencement of mental health conditions relative to each other.

Reviewers' comments:

Figure Resubmissions:

---

## [Editor Report · Decision Letter 3]

29 Sep 2025

Exploring the impact of drug decriminalization and legalization policies on mental health outcomes: A scoping review

PMEN-D-25-00024R3

Dear Dr Mohebbian,

We are pleased to inform you that your manuscript 'Exploring the impact of drug decriminalization and legalization policies on mental health outcomes: A scoping review' has been provisionally accepted for publication in PLOS Mental Health.

Best regards,

Craig Nicholas Cumming

Academic Editor

PLOS Mental Health